# A Systematic Review of Educational Interventions for Informal Caregivers of People Living with Dementia in Low and Middle-Income Countries

**DOI:** 10.3390/bs14030177

**Published:** 2024-02-24

**Authors:** Isabelle Evans, Ria Patel, Charlotte R. Stoner, Mel Melville, Aimee Spector

**Affiliations:** 1Faculty of Brain Sciences, Division of Psychology and Language Sciences, Research Department of Clinical, Educational and Health Psychology, University College London, Gower Street, London WC1E 6BT, UK; 2Centre for Chronic Illness and Ageing, Institute of Life Course Development, School of Human Sciences, University of Greenwich, Old Royal Naval College, Park Row, London SE10 9LS, UK; 3Research and Development Department, 1st Floor Maggie Lilley Suite, Goodmayes Hospital, North East London Foundation Trust, Barley Lane, Ilford, Essex, London IG3 8XJ, UK

**Keywords:** dementia, caregivers, intervention, education, developing countries, international

## Abstract

**Objectives:** With the increasing prevalence of dementia worldwide, there is a growing need for an integrated approach to dementia care. Little is known at present about the benefits of educational interventions for informal caregivers of people living with dementia (PLWD) in low- and middle-income countries (LMICs). This review aimed to identify and synthesise the current research on these interventions. **Method:** Four databases (PsycINFO, Medline, Web of Sciences and Scopus) were searched, alongside Google Scholar and reference lists. The Downs and Black checklist was used for quality assessment and data relating to intervention characteristics, outcomes, and educational component features were compared. **Results:** Eighteen papers detailing 17 studies were included. All studies presented found at least one significant outcome/effect. Study comparison was difficult due to diverse methodologies, intervention structures, and outcomes. Study quality was also variable. Four studies had education as the primary focus, and most interventions utilised multicomponent and group-based designs. Interventions that included group delivery tended to find more significant results than individual approaches. Intervention length did not appear to influence efficacy. Regular delivery and an average intervention dosage of around 12 h appeared most effective. **Conclusions:** Research into educational interventions for caregivers in LMICs appears to be promising and can help guide future interventions towards clinical implementation. A multicomponent group intervention trialled in Egypt provided particularly favourable findings. Future studies should focus on understanding the active mechanisms within such interventions to optimize their effectiveness. Collaboration between LMICs, high-income countries (HICs), and caregivers is crucial in developing interventions tailored to meet caregiver needs whilst accounting for feasibility and equity for dementia care worldwide.

## 1. Introduction

It is estimated that over 50 million people globally are currently living with dementia [1] and this figure is set to continue to rise. Reports estimate that over 74 million people will be diagnosed by 2030, with 63% of these individuals living in low- and middle-income countries (LMICs) [2]. These statistics emphasise the need for concerted efforts in formulating integrated approaches within dementia care and management, with consideration given to the diverse needs of different countries, cultures, and communities [3].

Many people living with dementia across the world are cared for and supported by informal or family caregivers [4]. For example, within the United Kingdom (UK) alone, informal caregivers contribute the equivalent economic value of £11.6 billion providing over 1.34 billion hours of unpaid care to people living with dementia [5]. Data regarding the frequency and economic cost of informal caregiving in LMICs are somewhat lacking in comparison to HICs [6,7]. However, some evidence suggests that familial caregivers are responsible for the majority of care [8], particularly in LMICs where formal dementia care policies or public health initiatives are often lacking [6,9]. High-income countries (HICs) have started to develop and implement diagnosis and treatment policies [10,11], such as National Dementia Plans [1], which will increase awareness and support, with further work needed. 

Research indicates that the caregiving role significantly affects health and wellbeing [12], encompassing increases in stigma [13], burden [14] and mental health difficulties [15]. Of note, many of the studies investigating these impacts are based on HIC populations. More recent research, however, suggests that the negative impact of caregiving is even greater for those from LMICs due to the lack of public health infrastructure for dementia [6,9]. In terms of stigma, studies have found both enacted and implied stigma within communities in Nigeria and South Africa [16,17] and this research suggests that the dearth of educational support for caregivers and communities allows such stigmas to propagate. Alongside this, Changoor [18] argues that the burden of dementia is greater in LMICs, supported by studies which detail an “amplified” burden on caregivers [19]. This amplification may be in part due to cultural narratives of collectivism and family involvement meaning family members are both assumed to take on caregiving roles and reluctant to ask for professional help for fear of social judgment [20], in addition to reduced public awareness and education [21]. Furthermore, research has found that distress is higher in caregivers in LMICs compared with HICs [22] due to “hardship” and “desperation” [19] which Wang et al. [9] suggested may be due to a lack of awareness about dementia as a neurodegenerative disorder rather than a normal part of ageing. This is mirrored in qualitative feedback from caregivers in LMICs around the lack of and need for psychoeducational interventions for caregivers [23,24]. 

Given the significant role and impact on informal caregivers, there is a large body of research into possible interventions to support them. A recent review [25] detailed the different categories of interventions that have been trialled including psychoeducation, counselling and psychotherapy, multi-component, and mindfulness-based interventions. A substantial number of studies highlight the value of interventions based on education around dementia, which can lead to reduced anxiety and depression symptoms, reduced burden, and increased quality of life for caregivers [26] as well as caregivers reporting an enhancement in their ability to care due to a better understanding of the illnesses [27,28]. These findings corroborate the aforementioned research emphasising the importance of public education and awareness of dementia in mitigating the impact of the caregiving role. Furthermore, the World Alzheimer’s Report [1] endorsed targeted public health campaigns worldwide as its first recommendation. An educational focus also limits the resources and timescale needed for the interventions, especially when they are delivered online, which then makes them a more cost-effective option for governments to consider implementing [29]. This is especially relevant for LMICs with limited funding [30], although it must be considered that computer/internet access may be challenging, and caregivers may lack the digital skills required. 

Despite the clear need for and likely benefits of educational interventions for informal caregivers, the majority of studies reviewed have been carried out in HICs [26,31]. Recent reviews of caregiver intervention studies in China and Asia found that over half of the studies had been carried out in HICs in the region [32,33] and a systematic review of culturally adapted interventions featured a large proportion of studies delivered in HICs to minority groups rather than in LMICs themselves [34]. There are no systematic reviews, therefore, that have considered caregiver interventions with educational components tested solely in LMICs. This review intends to address this gap in the literature, based on the following research aims:-To identify interventions with an educational component for informal dementia caregivers in LMICs;-To appraise the potential effects and quality of these intervention studies;-To make recommendations about future research regarding educational interventions for informal dementia caregivers in LMICs.

## 2. Method

### 2.1. Literature Search Strategy

This systematic review was performed in accordance with the Preferred Reporting Items for Systematic Reviews and Meta-Analyses (PRISMA) guidelines. The systematic literature search was conducted across four databases: PsycINFO, Medline, Web of Sciences, and Scopus After identifying pertinent papers from the initial search, a systematic reference check was conducted by reviewing the reference lists of included studies to ensure comprehensive coverage of relevant literature. No publication restrictions were imposed in terms of date; the search was conducted on 7 November 2022, and any literature published on or before that date was included. The search terms used were developed based on terms used in similar systematic review [35]. These were then tailored and refined through an iterative search process to enhance specificity for caregivers. Google Scholar was also searched to capture any literature that was not found via the database searches. 

Examples of search terms used to identify the population of people living with dementia included “Dementia” or “Alzheimer*” or “cognitive impairment” alongside “carer*” or “caregiver”. A list of LMIC search terms developed by the Cochrane Groups for CENTRAL [36], was used and included examples such as “low income countr*” or “underserved nation” or “Africa” or “Latin America” or “democratic republic of the congo” or “congo”. Search terms for the interventions included “support” or “training” or “intervention” or “course” or “education” or “awareness”. These examples can be found within the exhaustive list of search terms (see Appendix A).

Classification of HICs and LMICs was designated according to the World Bank 2022 [37]. The World Bank classifies countries based on Gross National Income (GNI) per capita. Countries with a GNI per capita above a certain threshold are classified as HICs, while those below the threshold are categorised as LMICs. In some cases, countries are categorised as LMICs overall, but there are specific areas or regions within these countries that are classified as HICs. For example, China is currently recognised as a LMIC, but Hong Kong, a special administrative region, is classified as a HIC. In instances where there is such differentiation within a country, the specific region where the study was carried out was screened to determine eligibility for inclusion in the study.

### 2.2. Inclusion and Exclusion Criteria

Studies were included if:They reported on an intervention that included a focus or component described as “educational”, “psychoeducational” or similar;The intervention, including the educational component, was evaluated using formal research methodology (quantitative or qualitative);The study population comprised of informal caregivers of people living with a diagnosis of dementia in the community;The intervention was delivered in a country (or region of a country) categorised as being low- or middle-income [37].

Studies were excluded if:They were not peer-reviewed;They were reviews or protocols;The intervention was also or solely delivered to the person living with dementia (PLWD);The intervention was delivered in a high-income country or region;The paper was not written in or translated into English;The intervention was solely delivered to professional caregivers;The full text was not available.

### 2.3. Quality Appraisal

Given the limited research within this area and the resulting number of papers identified, formal quality appraisal of the studies was carried out for evaluation rather than to a establish quality threshold for inclusion in the review. Duplicates were first removed using Endnote and then checked manually for accuracy. As not all the studies included were randomised control trials (RCTs), quality was assessed using The Downs and Black Checklist. The checklist comprises of 27 items and is designed to assess a range of different study designs [38]. Given that a considerable number of studies included in the review were either pilot studies or underpowered, the final checklist item—“did the study have sufficient power to detect a clinically important effect where the probability value for a difference being due to chance is less than 5%”—was modified to allow scoring options of Yes, No, or Unable to determine. Each study was scored on a scale of 0 to 27, where 27 represents the best possible score and 0 indicates the worst. Another researcher also independently scored 10% of the included studies using the checklist and a *k* statistic was calculated as 63.4% indicating a substantial level of agreement for interrater reliability [39]. Following this, the researchers discussed their differences and scored another paper independently achieving a *k* statistic of 100%. 

### 2.4. Intervention Evaluation 

All studies were evaluated in terms of general study characteristics. However, as the primary focus of this review was education, an in-depth evaluation was carried out for these components. This included considering the nature of the overarching intervention, the nature of the education delivery and the types of educational content. The thematic categories of these areas were decided by the researcher during the evaluation process. This process was informed by other reviews in the area that had made similar classifications ([4,25]). 

## 3. Results

Figure 1 provides an illustrative flow-chart of the literature screening process, depicting the inclusion of 18 papers corresponding to 17 distinct studies within this review. Two of the included papers stem from the same study, and two other papers represent separate studies but focus on the same intervention. Thus, there are 16 unique interventions, constituting 17 studies and encompassing 18 papers. 

### 3.1. Overview of Studies

Table 1 provides an outline of the 18 included papers, detailing 17 studies, including an overview of the study location, design, sample size, intervention details, outcome measures, significant results, and quality score. 

Studies were implemented across 11 different countries that covered four continents: South America, Asia, Europe, and Africa. All the papers were published between 2004 and 2022 and the samples ranged from 16 to 159 with an average sample size of 65. All the studies evaluated the interventions using quantitative outcome measures, there were no qualitative evaluations. There was a wide range of outcomes measured across the papers. In total, 36 different measures were used for caregiver outcomes and eight measures were used for people living with dementia. All the studies found at least one significant result.

### 3.2. Quality Appraisal

Details of the quality appraisal can be found in the Appendix A. The quality assessment ratings ranged from 13 to 25 out of 27. The average score was 20.72. Among the studies, the educative support group in Iran [46] received the lowest score of 13 out of 27. Conversely, the highest quality studies, scoring 25 out of 27, were part of an international research project on interventions for dementia caregivers, the 10/66 helping carers to care intervention in Russia and Peru [43,44]. Of the studies that scored more poorly, the most common reasons were no randomisation or blinding, a lack of clarity on intervention adherence, and the study not being sufficiently powered to detect a clinically significant effect. None of the studies reported on or measured any adverse events resulting from the intervention. 

### 3.3. Interventions

Table 2 summarises the interventions’ overall design, educational delivery methods and included educational content.

#### 3.3.1. Educational

Four interventions positioned education as their primary focus [46,47,50,53]. All the interventions were delivered in person to groups of caregivers. The psychoeducation group [50] followed the STAR-Caregivers model [58] which was a program initially designed and delivered in the USA, where all the other interventions appear to have been independently designed for LMIC populations. All the studies found at least one significant result relating to caregiver mental health or perceived burden. The educative support group study [46] found significant results on all the measured outcomes suggesting it is an efficacious intervention. However, the study quality was scored at 13/27, with no control group and a biased sample of only female caregivers which brings into question the validity of these results. Overall, the studies for the educational interventions were scored generally lower in terms of quality and only one [53] had a control group and only one measured intervention adherence [50] making it hard to draw firm conclusions on the effectiveness of the interventions. 

#### 3.3.2. Therapeutic

Two interventions detailed therapeutic delivery as their focus [42,51]. Both interventions were delivered in person to groups, with adherence to the intervention measured prior to analysis. Significant improvements in caregiver mental health were found in both studies. It is worth noting that in the counselling intervention [51], the improvement in caregiver mental health was found over time but not between the groups, control and experimental, suggesting that the intervention may not have been a relevant factor in the changes. The Cognitive Behavioural Therapy (CBT) intervention program [42] was the only study that found a significant result for a person living with dementia outcome across the whole review, in this case quality of life. The quality of the studies was mixed, with each scoring 22 and 17 [42,51]. The CBT program did not utilise a control group meaning the significant results cannot necessarily be attributed to the intervention.

#### 3.3.3. Support

Two interventions had an overarching focus on support [45,55]. Both interventions were delivered solely within group formats, one was delivered in person and the other via an online forum. The cultural-based support group [55] was based on interventions originally trialled in HICs where the professional facilitated peer support was tailored to delivery in China [45]. The online peer support study found significant results for four out of six measured outcomes relating to mental health and self-efficacy where the support group found one significant result relating to caregiver strain. Neither study utilised a control group and only the cultural-based support group [55] measured adherence, requiring at least 70% intervention completion. These design limitations are mirrored in the quality scores for both studies, 17 and 19 [45,55]. It is also worth considering that although a high level of engagement for the online peer support form was documented, 85% of participants having reviewed at least 75% of the information on the platform, it cannot be guaranteed how much of this information was read comprehensively. These limitations again mean it is not advisable for the significant results to be taken as definite evidence in favour of these interventions. 

#### 3.3.4. Multicomponent 

Nine interventions incorporated a range of different components ([32,40,41,43,44,48,49,52,54,56]). These included education, peer support, relaxation techniques, assessment, cognitive strategies, and mood management among other specific elements. Delivery methods varied, four interventions were delivered solely individually, via home visits for three and via an online platform for one. The other five interventions were delivered either solely via group in two studies, or via a mixture of group and individual for the remaining three interventions. The majority of the multicomponent interventions had a general focus but three of the interventions had more specific focusses such as caregiver self-management [56] or enhancing the positive aspects of caregiving [49]. Three of the interventions formed part of the wider 10/66 research program based on the “home care program” and “Helping carers to care” initiative [41,43,44]. The REACH VN multicomponent intervention [32,54] was also drawn from a wider research program, REACH VA [57] which was initially designed for delivery in the USA.

All the studies found significant results. The multicomponent psychosocial intervention program [52] found significant results for all the four measured outcomes relating to caregiver mental health, knowledge and burden. The other studies all found one or two significant results on a range of outcome measures from caregiver health to quality of life. 

The study quality for the multicomponent interventions was higher than the other categories, ranging from 20 to 25 out of 27. All of the studies included control groups, however three of the control groups used received a different form of active intervention ([32,40,56]) which could have interfered with the results. Three of the multicomponent interventions were the only studies to also consider retention and recruitment rates [32,40,56] within the review. Alongside this, another three studies were the only investigations to also consider follow-up of between four- and eight-weeks post-intervention completion [48,49,52]. Despite this increased quality of study design, only four of the studies measured intervention adherence, through number of online lessons completed [40] or attendance to sessions [32,41,49]. This again makes it difficult to draw conclusions on the effectiveness of the interventions, given that participants may not have completed the full intervention.

### 3.4. Intervention Delivery Characteristics 

#### 3.4.1. Group vs. Individual

There was a variety of intervention delivery styles, with nine being delivered in group settings, four being delivered individually and four using a combination of both. For the interventions that utilised groups, group size ranged from four to 20 participants. Five papers did not document the group sizes and three only provided a participant range. An average group size, using absolute group sizes and group range means, was calculated as 7.7 participants per group. One study involved an online forum so technically included an overall group size of 159 but this was not included in the group size calculation due to the difference in delivery style. No clear benefit of a particular group size can be concluded from this review. 

In terms of significant results, individual interventions found these only in relation to caregiver perceived burden, approaches to dementia and physical health. In comparison, interventions that utilised group or both group and individual, found significant results relating to different areas of caregiver mental health (anxiety, depression, distress) alongside perceived burden, dementia knowledge, quality of life and other more specific outcomes (e.g., self-efficacy). The study results do not indicate any benefit of using a mixed delivery approach of both group and individual, over solely group delivery. 

#### 3.4.2. Intervention Length and Dosage

The overall duration of the interventions ranged from four weeks up to 36 weeks, with all the interventions having more than one session or meeting expected. The most common intervention length was eight weeks (6 studies). The length did not appear to impact how efficacious the intervention was meaning that shorter-term interventions of four to eight weeks were no less effective than those delivered over six months. The two studies that found significant results for all the measured outcomes [46,52] both utilised eight-week durations, suggesting this appears to be an efficacious intervention length. However, the studies were mixed in terms of their quality with one not involving a control group [46].

It is worth noting that the length of the sessions themselves varied between the interventions from 30 min to two hours. Within this the regularity of the sessions also varied, seven interventions reported weekly sessions, one intervention noted fortnightly meetings and one intervention reported sessions every six to eight weeks. Of the remaining interventions, two were delivered online so there was no set attendance or session length, and the other six interventions did not detail this information. The multicomponent psychosocial intervention program [52] which found significant results and had a study quality score of 24/27, employed sessions of 45–60 min in length. This suggests that shorter sessions are potentially just as, if not more effective, than the longer sessions of two hours. 

Intervention dosage ranged from 2.5 to 39 h. Two interventions did not detail the session lengths to calculate dosage and two interventions were delivered online meaning participants had constant access for the study duration so dosage could not be quantified. Of the studies where dosage could be calculated, the average was 11.4 h. Most interventions used weekly sessions or visits. One intervention was delivered every 6 to 8 weeks [51] and no difference was found between the control and experimental groups in terms of caregiver mental health. This suggests that this irregularity of delivery could potentially be less effective than regular weekly delivery. The results also suggest that lowest dosage of 2.5 h may not be as efficacious, however it is worth noting that the intervention that delivered this dose [43,44] was also delivered to individuals, rather than in a group-based format, which may have contributed to less significant findings. The 39-h dosage intervention [50] was an outlier in terms of intensity and the caregiver outcomes did not indicate this was any more effective than an intervention dosage closer to the average.

#### 3.4.3. Internet-Based vs. In-Person

The delivery method utilised varied across the studies. Six of the studies were delivered face-to-face in settings such as hospitals or university buildings. Another five studies did not specify the location, but it is assumed by the nature of the intervention, the location, and the year of publication that these interventions also occurred face-to-face in public venues. Of the remaining studies, four were delivered through home-visits and two were delivered online. 

The two interventions that were delivered online, differed in terms of delivery, with one utilising professional facilitators and a community group [45] where the other provided an online learning platform where participants self-administered lessons [40]. The online iSupport program [40] found low levels of recruitment and retention documented as 44.67% and 36.42% respectively. It was also calculated that 31% of caregivers completed the recommended five or more lessons, and 45% did not complete any lessons at all. In comparison, the professionally facilitated online peer support found that 85% of participants reviewed at least 75% of the information and informal feedback also found that 92.4% of the caregivers thought the level of support received was important or very important. Alongside this, the peer support forum study found four significant results relating to caregiver mental health and self-efficacy, where the learning platform study only found a significant result relating to caregiver approaches to dementia despite also measuring mental health and self-efficacy outcomes. These differences suggest that online delivery may be more feasible and acceptable to caregivers in LMICs but that the presence of professional facilitators and peer interactions [45] may positively influence intervention adherence and effectiveness.

### 3.5. Educational Component

Educational delivery was divided into five categories: didactic, written, discussion, interactive, and individualised. Only one study utilised only one form of delivery and only one utilised all five. The most used were didactic and discussion-based delivery. Educational content was divided into five categories: dementia knowledge, delivering care, behavioural and psychological symptoms of dementia (BPSD), self-care, and local resources. Only one study delivered content on all five areas and all other studies delivered different combinations of two to four content areas. 

## 4. Discussion

### 4.1. Summary

Eighteen papers, detailing 17 different studies from 11 LMICs, were identified for inclusion within the review. It is somewhat challenging to directly compare the studies and the included interventions due to the differences in designs and measured outcomes, but the explorations within this review have resulted in a few key findings. 

Of the 16 different interventions, only four were categorised as having education as their primary focus, where the other 12 either incorporated multiple components including education or had education as a secondary component with a primary focus of peer support or therapy. This highlights the paucity of research in LMICs into caregiver education as a singular intervention component. However, all the studies found at least one significant result indicating that these interventions as a whole benefit caregivers and are a worthwhile research and public health avenue to pursue.

The studies that investigated interventions that utilised group delivery tended to find more significant results than the interventions that were delivered individually, particularly in relationship to caregiver mental health outcomes. This finding is supported by previous research that highlighted the value of peer support in wellbeing for caregivers [59]. 

Alongside this, the interventions did not appear to become more efficacious as their length increased meaning that shorter term interventions of four to eight weeks were no less effective than those delivered over six months. Similarly, shorter session lengths of 45–60 min seemed to be just as beneficial, if not more so, than longer sessions of two hours. In terms of overall intervention dosage, regular weekly or biweekly sessions accumulating to a total average of around 12 h looked to be the best fit for caregivers and their outcomes. These are important findings to consider given that there is often a lack of public funding and infrastructure in LMICs [9] that may prevent more rigorous and long-term interventions being implemented. Thus, interventions could be shorter and less intensive and benefit caregivers to the same degree. This may be particularly relevant given the evidence of lower public awareness and understanding of dementia in LMICS [60] which could mean short interventions, focussing on education, may have greater impact on outcomes than in HICs. 

In terms of online delivery, only two of the included studies utilised this approach. This demonstrates that research into this form of intervention is in its infancy within LMICs. The results of these studies showed potential promise for this delivery method but highlighted the need for professional facilitators and peer support to make it most effective. As such, it is possible online delivery that closely mimics an in-person group can capture the effectiveness of such interventions whilst also allowing for the benefits of using an online platform. 

Overall, the multicomponent psychosocial intervention program trialled by Shata et al. [52] provides the best example available at present of a high-quality study with promising results in terms of caregiver mental health, burden, and knowledge outcomes. The intervention incorporated all the components that have been shown in this review to be most effective; regular group sessions of around one hour delivered over eight weeks. Further research would be needed to support this conclusion especially considering intervention adherence and feasibility.

Of the 18 studies, only seven reported on intervention adherence, only three reported on feasibility in terms of recruitment and retention statistics and only three considered outcomes follow-up. These points highlight crucial methodological issues as most of these studies were initial investigations. As such, it is important to provide justification for more comprehensive evaluations in terms of clear conclusions and longstanding outcomes [61] but also in terms of participant endorsement [62]. In addition, none of the studies considered the cost of the intervention delivery which could also be critical in terms of feasibility when delivering in LMIC contexts with underdeveloped and underfunded services. This is especially important given that although many of these investigations were carried out over five years ago, no further comprehensive investigations or public health implementation of the interventions appear to have taken place. Attention needs to be paid to what is preventing this transition from research to public implementation and whether this is due to the interventions perhaps not being feasible or suitable for widespread delivery in LMICs.

### 4.2. Nature of the Interventions

Within the studies evaluated, researchers frequently explore diverse intervention styles, including educational, psychotherapeutic, and multicomponent approaches [25]. This is challenging for research clarity as these categorisations are arguably ambiguous and rely on subjective researcher decisions especially if limited information is provided on the intervention contents [63]; this hampers our understanding of the effectiveness of various intervention styles and their specific components. Researchers make recommendations about delivery methods, for example advising the use of peer groups to increase effectiveness in terms of caregiver psychological wellbeing [59]. However, there is a lack of exploration into the underlying active mechanisms or core components that result in positive outcomes for caregivers and attention needs to be given to this so that interventions contain the necessary components and avoid any that are redundant.

The variety of interventions being trialled, as captured within this review but also more widely within the field, perhaps demonstrates the diverse needs of caregivers but also the diversity of contexts where delivery is occurring. This underlines the need for research to not only consider the active mechanisms but also focus on the practical implementation of interventions in a range of clinical settings and whether interventions and active mechanisms are universal or culturally specific.

### 4.3. Education Delivery and Content 

There was a range of different educational delivery methods and educational content delivered within the interventions. The studies lacked coherence in terms of which components were included, and it was not possible to draw conclusions within this review regarding which were most efficacious. Literature mimics this lack of clarity about the most effective means of education delivery with some reviews concluding that individualised support is better [61], whilst others highlight the importance of group involvement [63]. There appears to be an overarching lack of research into the educational content being delivered and how this can be categorised. This may also be limited by the lack of consistency in how studies report on intervention contents, with some not giving any details and others providing manuals for replications. 

Attention needs to be given to how different researchers and interventions are conceptualising ‘education’. Different studies label this as “education”, “psychoeducation” or “training” for example, with little discussion in the literature about whether these labels capture the same concept. The lack of clarity around this issue can also be seen in the wide range of outcome measures used. There are unanswered questions at present as to why interventions include education if the primary focus is on other areas such as quality of life or burden, rather than knowledge. Research to understand the active mechanisms and content and how these impact different outcomes is needed to provide evidence-based rationales for the inclusion of education. 

### 4.4. Clinical Implications and Future Research

Moving forwards, therefore, there is a need for more high-quality research, such as RCTs, exploring dementia caregiver interventions in LMICs particularly in terms of dissecting the active components in terms of intervention design as well as with regards to educational delivery and content. This will allow for evidence-based designs and rationales for interventions. Assumptions about effective interventions must be avoided when drawn from research in HICs until there is an evidence base to support universal active mechanisms of change. There has been a move towards simultaneously developing interventions for both LMICS and HICs [64] as this could allow for designs suitable for widespread dissemination.

Further to this, the feasibility of the interventions also needs to be considered for the research to progress from academic to clinical implementation. Feasibility should capture recruitment and retention rates but also consider cost-effectiveness of the interventions as this is often neglected [65]. In terms of online intervention delivery, whilst in its infancy in LMICS, feasibility should capture access to technology and availability of digital skills among caregivers. A culturally adaptable Dementia Awareness for Caregivers course template was recently designed [66] that can be delivered to caregivers in LMICs in a one half-day session. Although yet to be formally evaluated, this study provides one of the first examples of a brief intervention for caregivers, which may be more easily disseminated into public health services than the more intensive interventions evaluated in this review. It also provides an example of an international template that can be adapted for different cultures and populations, where there is potential for both universal and culturally specific active mechanisms to be included when they become more clearly understood. 

Feasibility should also be considered in terms of participant endorsement and intervention acceptability. None of the present studies included qualitative evaluations. Qualitative data would allow researchers to understand how caregivers experience interventions, which designs, delivery methods and contents feel most helpful. Capturing caregiver opinions and priorities for interventions and outcomes could show how they align and diverge from other participant populations but also from researchers. This would thus allow for more co-produced and adaptable interventions, guided by universal but also culturally specific needs.

### 4.5. Limitations

This systematic review was not pre-registered on PROSPERO which increased the risk of similar reviews being conducted in parallel. Regrettably, PROSPERO does not permit retrospective registration, and this is recognized as a limitation in the study. Due to the small number of studies meeting the inclusion criteria (18 studies), all were included regardless of research quality; however, among these 18 studies, only 16 interventions were evaluated. Building on this, the Downs and Black quality appraisal checklist used does not capture all factors and excludes areas such as replicability and feasibility [38]. It is essential to recognize a methodological limitation, given that one researcher predominantly handled the selection process, screening of titles and abstracts, quality appraisal, and intervention evaluation.

The included studies also highlighted the lack of consensus in the research about the outcomes that interventions are expected to impact, with over 40 different outcome measures used. This means that direct comparison of study results was not possible. Consensus is needed within the field about the outcomes being considered, and those that are not, with rationales for this [25]. Ethically, this is also important as this lack of clarity often leads to poor prioritisation of measures meaning caregivers must complete more questionnaires unnecessarily [31].

The review included studies from across the world but only included reports written in English meaning that important findings from other cultures and settings may have been missed. Although the search strategy adopted was as comprehensive as possible, it cannot be discounted that grey literature was missed and other literature discounted due to its unavailability. 

Additionally, due to the variability in methods and findings, the review was unable to directly compare quantitative findings and as such, results were compared in terms of categories and thematic groupings chosen by the researcher which were reliant on papers reporting all included content. These groupings were based on somewhat subjective choices with Walter and Pinquart [63] arguing that such categorisations can be ambiguous. However, this process was informed by other reviews in the area that had made similar classifications ([4,25]).

## 5. Conclusions

There is no doubt that interventions for caregivers of people living with dementia in LMICs are needed, and that this need will continue to grow. This review indicates that the inclusion of educational content, delivered regularly within group settings over shorter times frames, in shorter sessions, is promising for caregiver interventions, with a range of significant results found. At present, the multicomponent psychosocial intervention trialled by Shata et al. [52] provides the best example of this. This research is in its infancy and further high-quality investigations are needed. It is not possible at this stage to identify the active mechanisms or components in terms of the overall intervention design, the educational delivery methods or the educational content included. Consideration also needs to be given to how education is being conceptualised and measured, the rationale for its inclusion and whether there are universal or culturally specific caregiver needs and outcomes. 

The aim of all studies regarding informal caregiver interventions should always be for widespread evidence-based public health implementation which appears not to have been prioritised historically. As such, collaboration between HICs and LMICs, and between researchers and caregivers, is advisable to work towards worldwide health equity for dementia with the clinical realities of intervention delivery in terms of outcomes, cost, feasibility, and cultural acceptability placed at the forefront. 

## Figures and Tables

**Figure 1 behavsci-14-00177-f001:**
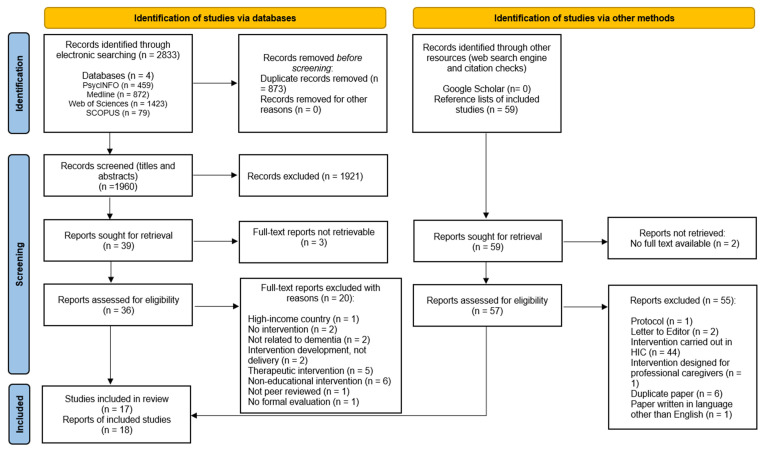
Flow diagram of literature identified, excluded, and included.

**Table 1 behavsci-14-00177-t001:** Overview of included studies.

Authors/ Date	Location/ Participants (*n*)/Duration	Design	Intervention	Dosage	Outcome Measures	Significant Results	Quality Score	Comments
Baruah, Varghese, Loganathan, Mehta, Gallagher-Thompson, Zandi, Dua and Pot, 2021 [40]	India *n* = 151 3 months	Randomised control trial Control: educational e-book Sample: National advertising and recruitment	Online iSupport Program 23 lessons related (with interactive learning situations) to themes: - What is dementia? (1) - Being a caregiver (4) - Caring for me (3) - Providing everyday care (5) - Dealing with behaviour changes (10) + Relaxation activity after each lesson	Online access for 3 months *N.B. Carers encouraged to attend 5+ lessons*	*Caregiver:* - Zarit Burden Interview (ZBI) - Center for Epidemiological Studies Depression–10 item scale (CES-D10) - Approaches to Dementia Questionnaire (ADQ) - RIS Eldercare Self-efficacy scale. - Mastery scale *PLWD:* None.	Significant difference in ADQ-19 scores (*p* = 0.030) at post-treatment between treatment and control—treatment had increase in positive attitudes towards PLWD.	**20/27**	Feasibility measured through recruitment and retention statistics.
Dias, Dewey, D’Souza, Dhume, Motghare, Shaji, Menon, Prince and Patel, 2008 [41]*	Goa, India *n* = 81 6 months *N.B.Visit frequency dependent on individual need.*	Randomised control trial Control: Waiting list Sample: Recruited through adverts and health services	Home Care Program Stepped care and tailored model. Delivered by home care advisors- - Basic education about dementia - Education about common behaviour problems and management - Caregiver support (e.g., in ADLs) - Referrals when behaviour problems escalated and needed medication intervention. - Networking of families to allow for support groups. - Advice regarding existing government schemes for elders	Average 9.225 h (Mean home visits = 12.3, average time = 45 min) *N.B. Nine additional peer support groups run*	*Caregiver:* -Neuropsychiatric Inventory Questionnaire (NPI-Q) Distress subscale D) - Zarit Burden Interview (ZBI) - General Health Questionnaire (GHQ) *PLWD:* -Neuropsychiatric Inventory Questionnaire (NPI-Q) (Severity subscale—S) - Everyday Abilities Scale for India (EASI)	Significant reduction in GHQ and NPI-Q (D) in the intervention group compared with control. (*p*-values not provided)	**23/27**	Dyad in study
Fialho, Köenig, Santos, Barbosa and Caramelli, 2012 [42]	Brazil *n* = 40 8 weeks	Quasi-experimental No control group. No sample or recruitment details	Cognitive-behavioural intervention program (Based on Training of Social Skills (TSS)) - Education - Cognitive, emotional, and social skills training - Support/empathy - Social comparison and shared learning - Strategies to modify own behaviour - Reinforcing persistence and effort - Cognitive strategies - Diary/therapy schedule - Activity organisation and preparation	16 h Weekly sessions (2 h)	*Caregiver:* - Zarit Burden Interview (ZBI) - Quality of Life scale for caregivers of people living with Alzheimer’s Disease (QoL-AD) - The list of stress symptoms (LSS) - Jalowiec Coping scale (JCS) - Trait Anxiety Scale (A-Trait) (from the State Trait anxiety Inventory (STAI)) - Major depressive episode module of the Mini International Neuropsychiatric Interview (NPI- Q MINI) 5.0. (DSM-IV). *PLWD:* -QoL-AD for PLWD (answered by family) - Mini mental state examination (MMSE), Disability assessment for dementia (DAD) and Neuropsychiatric Intervention Questionnaire (NPI-Q) (only pre-intervention)	Significant reduction in reported NPI-Q symptoms (*p* = 0.034) Significant reduction in trait anxiety scores (A scale—STAI) (*p* = 0.005) Significant improvement in PLWD QoL-AD (*p* = 0.040)	**17/27**	
Gavrilova, Ferri, Mikhaylova, Sokolova, Banerjee and Prince, 2009 [43] *	Russia *n* = 60 5 weeks	Randomised control trial (single blind parallel group) Control: Treatment as usual Sample recruited via medical centres	10/66 “Helping Carers to Care” Intervention 1—Assessment (1 session) (carer knowledge of dementia and family care arrangements) 2—Basic education (2 sessions) (introduction to dementia, the progression, causes, local care/treatment) 3—Training on ‘problem’ behaviour (2 sessions) (e.g., personal hygiene, dressing, repeated questioning, aggression, wandering)	2.5 h Weekly 30-min sessions	*Caregiver:* - Zarit Burden Interview (ZBI) - Self-reporting questionnaire (SRQ-20) - Caregiver quality of life (WHOQOL-BREF) -Neuropsychiatric Inventory Questionnaire (NPI-Q) *PLWD:* -Neuropsychiatric Inventory Questionnaire (NPI-Q) -DEMQOL	Significant reduction in burden (ZBI) for intervention group compared with control (*p* = 0.03)	**25/27**	
Guerra, Ferri, Fonseca, Banerjee and Prince, 2011 [44] *	Peru *n* = 58 5 weeks	Randomised control trial Control: Waiting list Sample: Local survey and memory clinic	10/66 “Helping Carers to Care” Intervention 1—Assessment (1 session) (carer knowledge of dementia and family care arrangements) 2—Basic education (2 sessions) (introduction to dementia, the progression, causes, local care/treatment) 3—Training on ‘problem’ behaviour (2 sessions) (e.g., personal hygiene, dressing, repeated questioning, aggression, wandering)	2.5 h Weekly 30-min sessions	*Caregiver:* - Zarit Burden Interview (ZBI) - Self-reporting questionnaire (SRQ-20) - Caregiver quality of life (WHOQOL-BREF) -Neuropsychiatric Inventory Questionnaire (NPI-Q) *PLWD:* -Neuropsychiatric Inventory Questionnaire (NPI-Q) -DEMQOL	Significant reduction in burden (ZBI) for intervention group compared with control (*p* < 0.001)	**25/27**	
Han, Guo and Hong, 2022 [45]	China *n* = 159 3–6 months *N.B. Carers could enter the intervention at different time points between 0 and 3 months*	Quasi-experimental—single group repeated measures. No control Snowballing sample via online forum and health clinics	WeChat virtual community—professional facilitated peer support 6 elements: 1—Peer emotional support 2—Lectures and consultation (13 topics—,e.g., dementia knowledge, care strategies, communicating) 3—Technique support 4—Reading 5—Maintaining a friendly environment 6—Participation and peer support	Online access for 3–6 months	*Caregiver:* - Self- Efficacy Questionnaire for Chinese Family Caregivers (SEQCFC) -Neuropsychiatric Inventory Questionnaire (NPIQ) - Perceived Stress Scale of Chinese version (PSS-C) - Center for Epidemiologic Studies Depression Scale (CES-D10) - Zarit Burden Interview (ZBI) - Learned Helplessness Scale *PLWD:* None.	Statistically significant decrease in stress (PSS-C) (*p* < 0.05), helplessness (*p* < 0.001) and depression (CES-D10) (*p* < 0.05) Statistically significant increase in self-efficacy (SEQCFC) (*p* < 0.05)	**17/27**	
Hinton, Nguyen, Nguyen, Harvey, Nichols, Martindale-Adams, Nguyen, Nguyen, Nguyen, Nguyen, Nguyen, Nguyen, Nguyen, Nguyen, Tiet, Nguyen, Nguyen, Nguyen, and Pham, 2020 [32] ***	Vietnam *n* = 60 2 to 3 months	Pilot cluster randomised control trial Control: Single 1:1 face to face educational session about dementia and written dementia resources Sample: Convenience through clustered local health services	REACH VN—manualised multicomponent intervention Home visits 4 core training sessions: 1—Problem solving 2—Mood management/cognitive restructuring 3—Stress management 4—Communication + 2 more session based on clinical judgment/caregiver needs	Estimated 8.6 to 13 h. Weekly 1-h home visits	*Caregiver:* - Zarit Burden Interview (ZBI) (4 item) - Patient Health Questionnaire (PHQ-4) - Alzheimer’s disease knowledge scale *PLWD:* None.	Significant decrease in burden in favour of intervention (ZBI) (*p* = 0.02) Significant decrease in PHQ-4 in intervention compared with control (*p* = 0.03)	**24/27**	Feasibility measured through recruitment and retention statistics.
Javadpour, Ahmadzadeh and Bahredar, 2009 [46]	Iran *n* = 29 8 weeks	Quasi-experimental repeated measures No control Random sample (no further details given)	Educative support group Each session contained: - 30-min educative talks providing information about dementia/challenging behaviours/problems faced by caregivers - 90-min interactive activities including discussions and sharing experiences	16 h Weekly 2-h sessions	*Caregiver:* - Perceived Stress Scale-10 (PSS-10) - General Health Questionnaire (GHQ) (Farsi) - Neuropsychiatry Inventory (NPI) *PLWD:* - Neuropsychiatry Inventory (NPI) - Clinical Dementia Rating (CDR)	Significant decreases in PSS scores (*p* = 0.0001), GHQ scores (*p* = 0.0001), NPI scores (*p* = 0.001)	**13/27**	All female caregivers
Kuzu, Beser, Zencir, Sahiner, Nesrin, Ahmet, Binali and Cagdas, 2005 [47]	Turkey *n* = 32 4 weeks	Quasi-experimental repeated measures No control Sample: Recruited through hospitals, Alzheimer’s association and community through word-of-mouth ad local media	Comprehensive educational program reinforced by an individualised component (CEPRIC) 3 components: 1—General information session (dementia, behaviour disorders, home and daily life) 2—Individualised educational component (specific problems identified through questionnaire) 3—Educational booklet	Not specified	*Caregiver:* - Duke Scale - Beck depression scale (BDS) - Beck anxiety inventory (BAI) *PLWD:* -Mini mental state examination (MMSE)	Significant decreases in BDS (*p* = 0.008), BAI (*p* = 0.01) Significant decreases in Duke scale subscales of physical health concerns (*p* = 0.001) and general health concerns (*p* = 0.004)	**18/27**	Dyad in study Nursing diagnoses also given before and after intervention
Magteppong and Yamarat, 2021 [48]	Thailand *n* = 60 8 weeks (follow-up at 20 weeks)	Quasi-experimental (pre/post parallel groups interventions study) Control: Treatment as usual (handbook provided post-intervention) Sample: Purposive via local hospital records and day centre attendees	Modified Transtheoretical Theory --of Stress and Coping (TTSC) Program (multicomponent) Aims: increase caregiver knowledge, reduce burden and increase quality of life Week: 1—Group health education (handbook provided) 2—Home visit (Stress, appraisal and coping) 3 -7—Telephone follow-ups 8—Home visit (Stress, appraisal and coping)	3.25—6.20 h Weekly contact- 1 × group meeting 2 × home visits 5 × telephone follow-ups	*Caregiver:* - Dementia Knowledge Assessment (DKA) - Thai burden interview for caregivers of patients with chronic illness - World Health Organisation’s Quality of life—Thai (WHO QoL) *PLWD:* None.	Significant increase in knowledge score for intervention compared with control at week 8 and 20 (*p* < 0.05) Significant difference in quality of life in favour of intervention compared with control at 8 and 20 weeks (*p* < 0.05)	**22/27**	
Pankong, Pothiban, Sucamvang and Khampolsiri, 2018 [49]	Thailand *n* = 72 8 weeks (follow-up at 12 and 20 weeks)	Randomised control trial Control: Treatment as usual Sample: Invited through local hospitals	Enhancing positive aspects of caregiving program 6 group sessions covering: 1—dementia knowledge/ADLs/behaviour management 2—meditation and spirituality 3—sharing experiences 4—role modelling/verbal reinforcements 1 individual session + dementia care booklet	12 h 6 × 2-h sessions *Additional phone call, length not specified*	*Caregiver:* - Positive aspects of caring questionnaire (PACQ) - Thai general wellbeing schedule (TGWS) *PLWD:* None.	Significant increase PACQ in intervention compared with control at weeks 8, 12 and 20 (*p* < 0.0001) Significant increase in wellbeing (TGWS) scores over time (*p* < 0.001) but no significant difference between the groups	**22/27**	
Santos, Sousa, Arcoverde and Dourado, 2013 [50]	Brazil *n* = 18 6 months	Quasi-experimental (pre/post) No control No sampling details given	Psychoeducational group (based on STAR-Caregivers model) Sessions included discussions about experiences, expressing emotions and educational lectures about dementia (types, BPSD ** etc.)	39 h Weekly 90-min sessions	*Caregiver:* - Caregivers version of quality of life in Alzheimer’s disease scale(QoL-AD) - Zarit Burden Interview (ZBI) - Beck depression inventory (BDI) - Beck Anxiety inventory (BAI) *PLWD:* -Clinical dementia rating (CDR) - Pfeffer Functional Activities Questionnaire (FAQ) -Cornell scale for depression in dementia (CSDD) -Neuropsychiatric Inventory Questionnaire (NPIQ)	Significant decrease in BDI scores between pre and post-assessments (*p* = 0.011)	**17/27**	
Senanarong, Jamjumras, Harmphadungkit, Klubwongs, Udomphanthurak, Poungvarin, Vannasaeng and Cummings, 2004 [51]	Thailand *n* = 50 6 months	Randomised parallel group intervention study. Control: Treatment as usual Sample: Recruited from hospital memory clinic	Counselling intervention for caregivers Content of group counselling and support sessions: - Sharing experiences - Information provided about techniques/coping - educational content (dementia prognosis and progression etc.) - Adaptions to environment - Identifying needs and understanding behaviours	3.75 h 5 × 45-min sessions (every 6–8 weeks)	*Caregiver:* Neuropsychiatric Inventory Questionnaire (NPIQ) *PLWD:* - Thai mental state examination (TMSE) - Functional assessment questionnaire (FAQ) -Thai activities of daily living measure -Clinical dementia rating (CDR)	Significant decrease in NPI-Q scores in intervention group between baseline and month 6 (*p* = 0.045) but not between the groups	**22/27**	
Shata, Amin, El-Kady and Abu-Nazel, 2017 [52]	Egypt *n* = 120 8 weeks—(post-measures after 3 months)	Randomised control trial Control: Waiting list Sample: Convenience sample through hospital clinic	Multicomponent psychosocial intervention program 3 components: 1- Group psychoeducation (2 sessions) 2—Brief group CBT (6 sessions) 3—Group support sessions (parallel to all sessions)	6–8 h Weekly 45–60-min sessions	*Caregiver:* - Knowledge questionnaire - Hamilton Depression Rating Scale (HDRS) (Arabic) - Taylor Manifest anxiety scale (TMAS) - Zarit Burden Interview (ZBI) *PLWD:* - Mini mental state examination (MMSE)	Significant decrease in anxiety (TMAS), depression (HDRS) and perceived burden (ZBI) for intervention compared with control at 8 weeks and 3 months (*p* < 0.001) Significant improvement in dementia knowledge in intervention group compared with control (*p* < 0.001)	**24/27**	
Tawfik, Sabry, Darwish, Mowafy and Soliman, 2021 [53]	Egypt *n* = 60 8 weeks	Randomised control trial Control: Treatment as usual Sample: Identified by researcher at Cairo University hospital outpatient unit	Psychoeducational Program Main objectives: 1—Giving information about different dementia behaviours (e.g., agitation, wandering) and tips to deal with them. 2—Caregiver support and de-stress techniques Sessions included role playing, brainstorming, group discussion and videos.	8 h Weekly 1-h sessions	*Caregiver:* - Zarit Burden Interview (ZBI) - Arabic Quality of life in Alzheimer’s disease questionnaire for caregivers (QoL-AD) *PLWD:* Arabic Quality of life in Alzheimer’s disease questionnaire for patients (QoL-AD)	Significant improvement in ZBI scores for intervention group compared with control at post-measure (*p* < 0.001)	**23/27**	
Tran, Nguyen, Pham, Nguyen, Nguyen, Nguyen, Harvey, and Hinton, 2022 [54] ***	Vietnam *n* = 60 2 to 3 months	Pilot cluster randomised control trial Control: Single 1:1 face to face educational session about dementia and written dementia resources Sample: Convenience through clustered local health services	REACH VN—manualised multicomponent intervention Home visits 4 core training sessions: 1—Problem solving 2—Mood management/cognitive restructuring 3—Stress management 4—Communication + 2 more session based on clinical judgment/caregiver needs	Estimated 8.6 to 13 h. Weekly 1-h home visits	*Caregiver:* REACH risk priority assessment (from REACH VA manual) (Variables: general health, caregiver frustrations, stress symptoms, general stress, behaviours, bother with behaviours) *PLWD:* None.	Significant decrease in caregiver frustration variable in intervention group compared with control (*p* = 0.01)	**22/27**	
Zakaria and Ab Razak, 2017 [55]	Malaysia *n* = 16 12 weeks	Quasi-experimental pre/post-study No control Sample: Convenience recruitment through a local memory clinic	Cultural-based support group Facilitated by healthcare professionals. Each session had 2 parts: 1—Psychoeducation session. 2—Mutual sharing and problem-solving Theme examples: - Introduction to principles and role within support group - understanding dementia - practical caregiving skills - supports for caregivers - effective communication - safe and healthy environment	12 h 2-h sessions every 2 weeks.	*Caregiver:* - Caregiver strain index (CSI) - Hospital anxiety and depression scale (HADS) - Caregiver quality of life (WHOQOL-BREF) *PLWD:* None.	Significant decrease in CSI scores from pre to post (*p* = 0.01) Significant improvement in specific domains of the WHOQOL-BREF from pre to post: physical (*p* = 0.01), psychological (*p* = 0.006) and environmental (*p* = 0.002)	**19/27**	
Zhang, Wu, Tang, Rong, Guo, Fang, Zhao, and Zhao, 2020 [56]	China *n* = 41 36 weeks	Quasi-experimental Control: individual telephone support Sample: Recruited from 2 hospitals	Caregiver self-management support intervention (C-SMS) Components: 1—Illustrated educational booklet (3 volumes—basic dementia care knowledge, symptom and problem identification and interventions, knowledge and skills for self-management) and a booklet of local contact details and support options) 2–6 bi-weekly support group sessions (12 weeks) 3–3 educational presentations during a 6-month follow-up period	15–18 h 6 × 2-weekly 2.5–3-h group sessions *+ 3 presentations with length not specified (over 6-month follow-up period)*	*Caregiver:* - Caregiver health related QoL (HRQoL) - Self-efficacy questionnaire for Chinese family caregivers (SEQCFC) *PLWD:* -Chinese version of the Disability Assessment in Dementia (DAD) -Neuropsychiatric Inventory-Questionnaire (NPI-Q)	Significant improvement in HRQoL in intervention compared with control (*p* = 0.017) Significant improvement in specific domains of self-efficacy for intervention compared with control: managing BPSD * (*p* = 0.013) and managing distress (*p* = 0.034)	**20/27**	Also measured physical outcomes—instances of caregiver metabolic syndrome Also measured retention and attrition statistics

* Three of the interventions formed part of the wider 10/66 research program based on the “home care program” and “Helping carers to care” initiative (Dias et al. [41]; Gavrilova et al., 2009 [43]; Guerra et al., 2011 [44]). ** BPSD: Behavioural and psychological symptoms of dementia. *** The REACH VN multicomponent intervention (Hinton et al., 2020 [32]; Tran et al., 2022 [54]) was drawn from a wider research program, REACH VA (Nichols et al., 2016) [57], which was initially designed for delivery in the USA.

**Table 2 behavsci-14-00177-t002:** Overview of intervention designs and education components.

Overall Study Content/Design	Study Authors	Educational Delivery Methods					Educational Content				
		Didactic	Written	Discussion	Interactive	Individual	Dementia Knowledge	Delivering Care	BPSD	Self-Care	Local Resources
Educational	*Javadpour et al., 2009 [46]*	**X**		**X**	**X**		**X**	**X**	**X**		
	*Kuzu, et al., 2005 [47]*	**X**	**X**	**X**		**X**	**X**	**X**	**X**	**X**	
	*Santos et al., 2013 [50]*	**X**		**X**	**X**		**X**	**X**	**X**		
	*Tawfik et al., 2021 [53]*	**X**		**X**	**X**				**X**	**X**	
Therapeutic	*Fialho et al., 2012 [42]*	**X**		**X**			**X**	**X**		**X**	
	*Senanarong et al., 2004 [51]*	**X**		**X**			**X**	**X**	**X**		
Support	*Han et al., 2022 [45]*	**X**		**X**	**X**		**X**	**X**	**X**	**X**	
	*Zakaria and Ab Razak, 2017 [55]*	**X**		**X**			**X**	**X**		**X**	
Multi-component	*Baruah et al., 2021 [40]*				**X**		**X**	**X**	**X**	**X**	
	*Dias et al., 2008 [41]*	**X**		**X**		**X**	**X**		**X**		**X**
	*Gavrilova et al., 2009 [43]/Guerra et al., 2011 [44]*	**X**				**X**	**X**		**X**		**X**
	*Hinton et al., 2020 [32]/Tran et al., 2022 [54]*	**X**	**X**			**X**		**X**	**X**	**X**	
	*Magteppong and Yamarat, 2021 [48]*	**X**	**X**	**X**		**X**	**X**	**X**	**X**	**X**	
	*Pankong et al., 2018 [49]*	**X**	**X**	**X**	**X**	**X**	**X**	**X**	**X**	**X**	
	*Shata et al., 2017 [52]*	**X**		**X**	**X**		**X**		**X**	**X**	
	*Zhang et al., 2020 [56]*	**X**	**X**	**X**			**X**	**X**	**X**	**X**	**X**

## Data Availability

No new data were created or analyzed in this study. Data sharing is not applicable to this article.

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
