# Peer review of "A Systematic Review of Educational Interventions for Informal Caregivers of People Living with Dementia in Low and Middle-Income Countries"

_behavsci, 2024, doi:10.3390/bs14030177_

Round 1
Reviewer 1 Report
Comments and Suggestions for Authors
Review Comments: A systematic review of educational interventions for information caregivers of people living with dementia in low and middle-income countries.
1. Abstract Line 34: You use HICs but did not identify this in the abstract. Please add this definition of the acronym
2. Lines 54-56: This sentence is confusing. It does not begin with a capital, and then two concepts are combined in the sentence. This may need to be two sentences.
3. Line 61: You indicate that research indicates a “significant impact” and then go on to identify stigma, etc. However, you do not identify what kind of impact you are addressing. Is it mental health impact, well-being, etc.?
4. Line 138 Section 2 Method: You indicated that the HIC and LMIC were defined using the World Bank Criteria, but it would be helpful to know that that is. Please include a short description of these criteria and the differences in designation.
5. I found all of the section 3’s difficult to navigate. You would have Results and then some short description of the following table or flowchart. This would flow better if you provided a short paragraph of that section, what it is, how to interpret the table or flowchart, and then add the chart or table as an Appendix.
6. Flowchart Figure 1: The flowchart does not specify why you moved to 16 from the 17 included studies for review. It is important to know why the number was moved to 16 in the discussion. You also discuss the #16 in limitations, but we don’t know why it moved from 18 to 16.
7. Line 13 in Section 3.2 Quality Appraisal: You introduce 10/66, but this has not been seen before, and I was unsure what it referred to. Please clarify.
Reviewer 2 Report
Comments and Suggestions for Authors
Dear authors
I think it is an interesting and useful study. The quality of the article is very good and it shows that the authors put a lot of effort into writing it.
I offer some small comments that I hope will be of interest to you.
- The abstract is well written. In the conclusion part, it is better to explain the clinical applications of the study. It is better to choose keywords from mesh.
Introduction: Well tuned. Browsing the texts is well done. Objectives are well defined.
Methodology: Very well written.
Findings: Figure 1 is a bit long and boring, it is better to make it shorter. Table number 1 is very long. It is better to be more concise. Table 2 should be deleted. The data presented in the table format do not need to be repeated in the text.
Discussion: Well written.
References: Some of the references are not written according to the journal format and should be edited.
With respect
Comments on the Quality of English Language
I think some small edits are needed.
